Phytochemical content, especially spermidine derivatives, presenting antioxidant and antilipoxygenase activities in Thai bee pollens

Khongkarat Phanthiwa 1
Phuwapraisirisan Preecha 2
Chanchao Chanpen chanpen.c@chula.ac.th 3
1 Program in Biotechnology, Faculty of Science, Chulalongkorn University , Patumwan , Bangkok , Thailand
2 Center of Excellence in Natural Products, Department of Chemistry, Faculty of Science, Chulalongkorn University , Patumwan , Bangkok , Thailand
3 Department of Biology, Faculty of Science, Chulalongkorn University , Patumwan , Bangkok , Thailand
Adnan Mohd
Electronic publication date: 2022 May 25
Publication date: 2022
Volume: 10
Electronic Location ID: e13506
Received 2022 Jan 31; Accepted 2022 May 6
Copyright: ©2022 Khongkarat et al.
Copyright year: 2022
Copyright holder: Khongkarat et al.
License: This is an open access article distributed under the terms of the Creative Commons Attribution License, which permits unrestricted use, distribution, reproduction and adaptation in any medium and for any purpose provided that it is properly attributed. For attribution, the original author(s), title, publication source (PeerJ) and either DOI or URL of the article must be cited.
License URL: https://creativecommons.org/licenses/by/4.0/

Keywords: Bee pollen, Bioactivity, Flavonoid content, Phytochemical, Total phenolic content

Funding: Science Achievement Scholarship of Thailand 90th Anniversary of Chulalongkorn University Fund (Ratchadaphiseksomphot Endowment Fund) Toray Science Foundation, and Thailand Science Research and Innovation Fund Chulalongkorn University CUFRB65_food(6)_114_23_44 This work was financially supported by the Science Achievement Scholarship of Thailand, the 90th Anniversary of Chulalongkorn University Fund (Ratchadaphiseksomphot Endowment Fund), Toray Science Foundation, and Thailand Science Research and Innovation Fund Chulalongkorn University (CUFRB65_food(6)_114_23_44). The funders had no role in study design, data collection and analysis, decision to publish, or preparation of the manuscript.

==============================
Background

Bee pollen (BP) is full of useful nutrients and phytochemicals.Its chemical components and bioactivities depend mainly on the type of floral pollen.

Methods

Monofloral BP from Camellia sinensis L., Mimosa diplotricha, Helianthus annuus L., Nelumbo nucifera, Xyris complanata, and Ageratum conyzoides were harvested. Crude extraction and partition were performed to yield solvent-partitioned extracts of each BP. Total phenolic content (TPC) was assayed by the Folin-Ciocalteu method, while the flavonoid content (FC) was measured by the aluminium chloride colorimetric method. Antioxidant capacity was measured by the (i) 1,1-diphenyl-2-picrylhydrazyl (DPPH) radical scavenging activity, (ii) 2,2’-azino-bis (3-ethylbenzthiazoline-6-sulphonic acid) (ABTS) scavenging activity and its Trolox equivalent antioxidant capacity (TEAC), and (iii) ferric reducing antioxidant power (FRAP). All samples were tested for lipoxygenase inhibitory (LOXI) activity. The most active sample was enriched by silica gel 60 column chromatography (SiG60-CC) and high performance liquid chromatography (HPLC), observing the chemical pattern of each fraction using thin layer chromatography. Chemical structure of the most active compound was analyzed by proton nuclear magnetic resonance and mass spectrometry.

Results

Dichloromethane (DCM)-partitioned BP extracts of H. annuus L. and M. diplotricha (DCMMBP) showed a very high TPC, while DCMMBP had the highest FC. In addition, DCMMBP had the strongest DPPH and ABTS radical scavenging activities (as a TEAC value), as well as FRAP value. Also, DCMMBP (60 µg/mL) gave the highest LOXI activity (78.60 ± 2.81%). Hence, DCMMBP was chosen for further enrichment by SiG60-CC and HPLC. Following this, the most active fraction showed higher antioxidant andLOXI activities with an EC50 for DPPH and ABTS of 54.66 ± 3.45 µg/mL and 24.56 ± 2.99 µg/mL (with a TEAC value of 2,529.69 ± 142.16 µmole TE/g), respectively, and a FRAP value of 3,466.17 ± 81.30 µmole Fe2+/g and an IC50 for LOXI activity of 12.11 ± 0.36 µg/mL. Triferuloyl spermidines were revealed to be the likely main active components.

Conclusions

TPC, FC, and spermidine derivatives played an important role in the antioxidant and antilipoxygenase activities in M. diplotricha bee pollen.

Introduction

People nowadays have a longer expected lifespan due to advances in medicine and technology. In addition, people pay more attention to their health. Hence, a lot of nutrient supplements have been introduced, which are mostly from natural products. Bee pollen (BP) is one of the natural bee products that is widely used for nutritional and medical applications. It is produced by foraging bees by mixing the floral pollens with some nectar or honey, enzymes, wax, and bee secretion (Chantarudee et al., 2012). Bee pollens are a good protein source because they contain a high level of well-balanced proteins (those with all the essential amino acids necessary for the human body). Furthermore, BPs contain other required nutrients, including unsaturated fatty acids, vitamins, minerals, and trace elements. As a result, it is commonly used as a natural dietary supplement (Sommano et al., 2020).

Additionally, BP contains a variety of beneficial chemicals, particularly phenolic and flavonoid components, such as quercetin, kaempferol, caffeic acid, and naringenin (Saric et al., 2009). As a result, many bioactivities of BP have been discovered, such as antimicrobial and anti-inflammatory activities (Saral et al., 2022). The phenolic compounds from rape BP showed in vitro antioxidant and tyrosinase inhibition (TYRI) properties and could inhibit melanogenesis. These activities were attributed to rutin, the major flavonoid in the extract (Sun et al., 2017). Also, the phenol and flavonoid compounds in BP from Nigeria had an α-amylase inhibitory activity, indicating the potential therapeutic potential of BP in the management of diabetes (Daudu, 2019). In addition, a monoamine oxidase (MAO) inhibitory activity was reported for chestnut BP (Yildiz et al., 2014), and so its potential use in the treatment of depressive disorders, Parkinson’s disease, and Alzheimer’s disease. The relationship between MAO inhibition and the total phenolic content (TPC) as well as the antioxidant capacity of BP was subsequently reported (Yildiz et al., 2014). Hence, the TPC and flavonoid contents (FC) from BP, which mainly depended on the floral origin, have the potential to be used as a natural antioxidant agent and enzyme inhibitor.

The antioxidant and lipoxygenase inhibition (LOXI) properties were found to be associated with anti-inflammatory, anti-cancer, and anti-aging effects (Eshwarappa et al., 2016). The antioxidant properties can protect cells from the oxidative damage of free radical molecules, which are unstable molecules with one or more unpaired electrons (Lobo et al., 2010). The antioxidant can react with other molecules, including the intracellular proteins, lipids, and DNA. After such damage, the cells are in an oxidative stress that can induce the development of many chronic diseases, such as cancer, autoimmune disorders, aging, cataracts, rheumatoid arthritis, and cardiovascular and neurodegenerative diseases (Pham-Huy, He & Pham-Huy, 2010).

Therefore, to help prevent oxidative damage to cells, antioxidant agents that can donate electrons or hydrogen atoms are important in order to reduce the free radical level (Shahidi & Zhong, 2015). The lipoxygenases (LOXs) are an important family of enzymes in inflammatory and immune responses. They catalyze the conversion of polyunsaturated fatty acids and arachidonic acid to inflammatory eicosanoids, including hydroperoxy-eicosatetraenoic acid and leukotrienes (Mashima & Okuyama, 2015). The overexpression of LOX promotes the development of several inflammation-related diseases, such as arthritis, asthma, cancer, and allergic diseases (Wang et al., 2021). This can be prevented via the inhibition of the LOX pathway and the targeting of LOX with LOXIs is a promising therapeutic target for treating a wide spectrum of human diseases (Eshwarappa et al., 2016).

The chemical composition of BP mainly depends on its floral and geological/ geographical origins. In Thailand, some dominant floral origins are different from those in other countries (Chamchumroon et al., 2017), while the bioactivities of some BPs have not been reported before. This led to our interest in the bioactivities of different types of Thai monofloral BPs. Six such samples were collected and then sequentially extracted and partitioned with three organic solvents of different polarities. The enrichment of the active compounds, using bioactivity and chemical profile screening of the fractions, was performed. The structure of the isolated active compounds was analyzed by proton nuclear magnetic resonance spectroscopy (1H-NMR) and the molecular weight (MW) was measured by MALDI-TOF mass spectrometry (MS). The benefit of this work is the discovery of potential new antioxidant and LOXI compounds and the promotion of BP as a nutraceutical food to be developed for use in the pharmaceutical industry. This may encourage the growth of Thai apiaries, resulting in an improved income for bee farmers.

Materials & Methods

Sample collection

Six monofloral Apis mellifera BP samples were collected from five provinces in Thailand (Table 1) and stored at room temperature (25 °C) until used. The BP was identified by palynological analysis and sequence analysis of the ITS-2 region of the rRNA genes, as previously reported (Khongkarat et al., 2022). Both analyses were consistent and confirmed that the six A. mellifera BP samples were from Camellia sinensis L. (BP1), Mimosa diplotricha (BP2) (Chiangmai province), Helianthus annuus L. (BP3) (Lopburi province), Nelumbo nucifera (BP4) (Nakhon Sawan province), Xyris complanata (BP5) (Udon Thani), and Ageratum conyzoides (BP6) (Lamphun).

Crude extraction and partition

All collected samples were extracted as previously described (Umthong et al., 2011). In brief, each BP sample (140 g) was extracted with 800 mL of methanol (MeOH) by shaking at 100 rpm, 15 °C, for 18 h. After that, the supernatant was collected and the residue was extracted two more times each with 800 mL of MeOH. The three supernatants were pooled and dried under reduced pressure to obtain the respective MeOH crude extract (MCE). Each MCE was further partitioned by hexane, dichloromethane (DCM), and MeOH as follows. At first, the MCE was dissolved in MeOH (250 mL) and then mixed with an equal volume of hexane in a separating funnel and left to phase separate. The upper hexane phase was collected and the lower MeOH phase was further partitioned two more times in the same manner, with the three hexane extracts being pooled and dried under reduced pressure to yield the respective hexane-partitioned BP extract (HX); designated as HXCBP, HXMBP, HXHBP, HXNBP, HXXBP, and HXABP for BP1–6, respectively. Next, the residual MeOH phase was added to an equal volume of water and subsequently partitioned three times with DCM in the same manner as above (except the DCM phase was the lower layer), with the pooled DCM extracts being evaporated as above to provide the DCM-partitioned extracts; designated as DCMCBP, DCMMBP, DCMHBP, DCMNBP, DCMXBP, and DCMABP for BP1–6, respectively. Finally, the residual MeOH phases were evaporated as above to obtain the MeOH-partitioned extracts; designated as MTCBP, DCMMBP, MTHBP, MTNBP, MTXBP, and MTABP for BP1–6, respectively. All partitioned extracts were kept at −20 °C in the dark until used.

Table 1 Detail of sample collection.

Type of BP	Sample code	Collecting time	Collecting site (province)	Geographical location	
				Longitude	Latitude	
Camellia sinensis L. BP	BP1	February, 2018	Chiangmai	99°03′45.3″E	18°46′37.3″N	
Mimosa diplotricha BP	BP2	February, 2018	Chiangmai	99°03′45.3″E	18°46′37.3″N	
Helianthus annuus L. BP	BP3	February, 2018	Lopburi	101°01′07.5″E	14°51′31.7″N	
Nelumbo nucifera BP	BP4	February, 2018	Nakhon Sawan	100°15′01.4″E	15°41′02.4″N	
Xyris complanata BP	BP5	February, 2018	Udon Thani	102°45′28.8″E	17°21′12.7″N	
Ageratum conyzoides BP	BP6	February, 2018	Lamphun	98°53′42.6″E	18°04′49.2″N	

Determination of the TPC

The TPC of each partitioned extract was evaluated as previously reported (Marghitas et al., 2009) based on the Folin-Ciocalteu method. The method was adapted to be performed in a 96 well plate. Firstly, 125 µL of 0.2 N Folin-Ciocalteu reagent was added to 25 µL of the diluted partitioned extract or gallic acid solution in dimethyl sulfoxide (DMSO) and mixed for 5 min. Then, 100 µL of 7.5% (w/v) sodium carbonate solution was added per well and incubated at room temperature for 2 h. The absorbance at a wavelength of 700 nm was measured using a microplate reader against DMSO as a blank. All reactions were performed in triplicate. The results were expressed as mg of gallic acid equivalents (GAE)/g of partitioned extract using a standard graph for gallic acid in the range of 0.02–0.1 mg/mL.

Determination of the FC

The FC of each partitioned extract was measured as previously described (Marghitas et al., 2009) based on the aluminium chloride (AlCl3) colorimetric method but with adaptation for use in a 96 well microplate reader. Briefly, 25 µL of the diluted partitioned extract solution dissolved in DMSO was mixed with 100 µL of distilled water. Then, 7.5 µL of 5% (w/v) sodium nitrite solution was added and incubated at room temperature for 5 min, followed by the addition of 7.5 µL of 10% (w/v) AlCl3. After 6 min of incubation at room temperature, 50 µL of 1 M sodium hydroxide was added and the absorbance at wavelength of 510 nm was measured. Each assay was performed in triplicate. The FC was expressed as mg of quercetin equivalents (QE)/g of partitioned extract using a standard graph for quercetin (0.1–1 mg/mL).

Determination of the antioxidant capacity

The DPPH free radical scavenging activity assay

The potential of the partitioned extracts was determined as previously described (Khongkarat et al., 2020). Five different concentrations of each sample were prepared in DMSO. For each concentration, 20 µL of the sample was mixed with 80 µL of 0.15 mM DPPH in MeOH and incubated at room temperature for 30 min. The absorbance was measured at a wavelength of 517 nm (A517) using a microplate reader. Ascorbic acid (vitamin C) was used as the standard reference. Each assay was performed in triplicate. The free radical scavenging activity was calculated from Eq. (1): (1) % DPPH radical scavenging activity=A−B/A×100,

where A is the A517of the negative control and B is the A517 of the treatment.

The % inhibition (Y axis) was plotted against the extract concentrations (X axis) and the effective concentration at 50% (EC50) was obtained from the graph.

Determination of the Trolox equivalent antioxidant capacity (TEAC)

The TEAC assay followed the described method (Suriyatem et al., 2017). The stock ABTS•+ solution was prepared by reacting 7 mM ABTS solution with 2.45 mM potassium persulphate solution in distilled water at a 1:1 (v/v) ratio in the dark at room temperature for 16 h before use. The working ABTS•+ solution was prepared by diluting the stock ABTS•+ solution (one mL) with ethanol (35 mL) to get an absorbance at 734 nm (A734) of 0.700 ± 0.025. After that, 0.3 mL of each partitioned extract at different concentrations dissolved in DMSO was mixed with 2.7 mL of the prepared ABTS•+ solution, left for 6 min in the dark and then the A734 was read. The percentage of inhibition was calculated according to Eq. (2), (2) % ABTS radical scavenging activity=A−B/A×100,

where A is the A734of the negative control and B is the A734of the treatment.

The % inhibition (Y axis) was plotted against the respective extract concentration (X axis) and the EC50 value was obtained from the graph. The results were then compared with the Trolox standard curve (0–0.2 mM) and the results are expressed as µmole Trolox equivalents (TE)/g partitioned extract.

Determination of the ferric reducing antioxidant power (FRAP)

The FRAP assay was performed as previously described (Sun et al., 2017). The FRAP reagent was freshly prepared by mixing 100 mL of 300 mM acetate buffer (pH 3.6), 10 mL of 10 mM 2,4,6-tris(2pyridyl)-s-triazine solution in 40 mM hydrochloric acid, and 10 mL of 20 mM ferric chloride solution. Later, 100 µL of the sample at a desired concentration was added to three mL of the FRAP reagent and incubated at 37 °C in a water bath for 6 min. The absorbance was measured at a wavelength of 593 nm. Aqueous solutions of ferrous sulfate (0–2,000 µM) were used to establish the standard graph and the reducing capacity was expressed as µmole of Fe2+/g extract. Each assay was performed in triplicate.

In vitro TYRI activity

The in vitro TYRI activity was determined as described (Khongkarat et al., 2020). The reaction mixture contained 120 µL of 2.5 mM L-DOPA in 80 mM phosphate buffer (pH 6.8), 30 µL of 80 mM phosphate buffer (pH 6.8), and 10 µL of partitioned extract or kojic acid (positive control) at different concentrations in DMSO. After mixing, the reaction was pre-incubated at 25 °C for 10 min. Then, 40 µL of 165 units (U)/mL mushroom TYR in phosphate buffer was added and incubated at 25 °C for 5 min. The absorbance at 475 nm (A475) was measured using a microplate reader. The TYRI activity was calculated as the IC50 value. Each assay was performed in triplicate and the percentage TYRI was calculated from Eq. (3); (3) Percentage of tyrosinase inhibition=A−B−C−D/A−B×100,

where: A is the A475 after incubation without extract, B is the A475 after incubation without an extract and tyrosinase, C is the A475 after incubation with an extract and tyrosinase, and D is the A475 after incubation with an extract, but without tyrosinase.

The % TYRI (Y axis) was plotted against the respective extract concentration (X axis) and the IC50 value was obtained from the graph.

In vitro LOXI activity

The LOXI activity of each partitioned extract was determined as described (Sethiya & Mishra, 2014) with some modifications. Firstly, 220 µL of 0.2 M borate buffer pH 9.0, 30 µL of the extract in DMSO, and 250 µL of 20,000 U/mL LOX from soybean in 0.2 M borate buffer pH 9.0 were mixed and incubated at 25 °C for 5 min. After that, 1,000 µL of 0.6 mM linoleic acid solution was added and the absorbance at 234 nm (A234) was measured. Nordihydroguaiaretic acid (NDGA) was used as the reference standard. Each assay was performed in triplicate. The percentage of LOXI was calculated from Eq. (4); (4) Percentage of LOXI=A−B−C−D/A−B×100,

where A is the A234 after incubation without an extract, B is the A234after incubation without an extract and LOX, C is the A234 after incubation with an extract and LOX, and D is the A234 after incubation with an extract but without LOX.

The % LOXI (Y axis) was plotted against the respective extract concentration (X axis) and the IC50 value was obtained from the graph.

Enrichment of active fractions

Among the mentioned bioactivities, the most active extract for antioxidants and in vitro LOXI activity was further fractionated following the bioactivity as in previous studies (Teerasripreecha et al., 2012).

Fractionation by silica gel 60 column chromatography (SiG60-CC; 500-mL size)

The most active extract for antioxidant and in vitro LOXI activities was selected for fractionation by SiG60-CC. Briefly, the 500-mL column was packed with fine SiG60 (Merck, for CC). The partitioned extract (5.64 g) was dissolved in 20 mL of MeOH and combined with 20 g of rough SiG60 (Merck, for CC). After drying, it was poured over the surface of the packed SiG60 column and then eluted with 6.5 L of DCM, 8.5 L of 7% (v/v) MeOH in DCM, and 3.5 L of MeOH, respectively. Eluted fractions (250 mL each) were collected, and the solvent was removed by evaporation under reduced pressure at a maximum temperature of 40–45 °C. The pattern of chemical compounds in each fraction was profiled by thin layer chromatography (TLC; see below). Fractions with the same TLC pattern were pooled together and tested for antioxidant and in vitro LOXI activities.

Chemical profiling by TLC

A sample was spotted onto the starting line of a 5 × five cm2 TLC plate (silica immobile phase) using a capillary tube and then dried at room temperature. It was then resolved in one direction using 7% (v/v) MeOH: DCM as the mobile phase. The resolved compounds on the TLC plate were visualized under UV light at 254 nm or by dipping in 3% (v/v) anisaldehyde in MeOH and heating over a hot plate.

Enrichment by HPLC

To separate (enrich) the compounds of a similar polarity in the mixture (extract) the HPLC method reported by Lv et al. (2015) was further developed and modified. The optimal operating condition was found using a SB-PHENYL column (5 µm, 9.4 × 250 mm), loading 10 × 20 µL aliquots of the respective sample (100 mg/mL in MeOH) with a column temperature of 25 °C, and eluting in an isocratic mobile phase (two mL/min) of milli Q H2O and MeOH ranging from 0:100 to 60:40 (v/v) H2O: MeOH. The eluted fractions were detected by UV-visible spectroscopy at 254 nm (A254). The retention time of the extract was determined.

Chemical structure analysis by 1H-NMR and MS

Among the selected fractions from the SiG60-CC (500 mL size) and HPLC fractions, the most active fraction for both activities was evaporated and analyzed as reported (Do et al., 2021). Briefly, the evaporated sample was dissolved in an appropriate deuterated solvent (methanol-d4, Merck) at a ratio of 5–20 mg of compound to 600 µL of the solvent. Next, it was transferred to an NMR tube and shaken until totally dissolved. The 1H-NMR spectrum was recorded using a Jeol JNM-ECZ 500MHz operated at 500 MHz for 1H-NMR nuclei with tetramethylsilane as the internal standard. The chemical shift in δ (ppm) was assigned with reference to the signal from the residual protons in the deuterated solvents, while the chemical shift and J coupling value were determined using the MestReNova version 12.0.3 software. The MW of the active fractions was analyzed using a microTOF focus II MS with electrospray ionization in the positive mode and α-cyano-4-hydroxycinnamic acid as the matrix.

Data analysis

All experiments were done in triplicate. Numerical data are reported as the mean ± one standard deviation (SD), determined in the Microsoft Excel 2019 software (Khongkarat et al., 2022). One-way ANOVA and T-test were used to test for significant differences. Tukey’s and Dunnett T3 test (p < 0.05) was applied for pairwise multiple comparisons (Durovic et al., 2022). All statistical analyses were performed using the IBM SPSS statistics version 22 for windows software.

The overall procedure of BP screening and enrichment of the antioxidant and enzyme inhibitory activities from the most active extract is summarized schematically in Fig. 1.

Figure 1 Summary of the extraction, screening, and enrichment procedures for the selected BP.

Results

The partitioned extracts of BPs

After partitioning the MCEs of the six different types of BPs with organic solvents from the lowest to the highest in polarity (hexane, DCM, and MeOH), a total of 18 partitioned extracts were obtained. These partitioned extracts were then evaporated and weighed, and the character of each extract was observed, with the results summarized in Supplement 1. The MeOH-partitioned extracts had the highest yield (above 40%). Only the DCM-partitioned extracts exhibited a sticky solid form, whereas the MeOH- and hexane-partitioned extracts exhibited an oil form. The TPC and FC of the MeOH-, DCM- and hexane-partitioned extracts of all six types of BP were then determined.

Determination of the TPC and FC

The TPC and FC were determined from the calibration curves of gallic acid (y = 8.5519x − 0.0133; R2 = 0.9991) and quercetin (y = 0.6342x + 0.0083; R2 = 0.9982), respectively, with the TPC and FC of each extract shown in Table 2. The effect of the partition solvent on the TPC and FC was significantly evident, with the highest TPC and FC being found in the DCM-partitioned extracts of all six BP samples, followed by the MeOH-partitioned extracts, while the hexane-partitioned extracts had the lowest TPC and no FC. The TPC of these BP extracts varied between 7.20 ± 0.25 and 53.26 ± 0.85 mg GAE/g, being highest in DCMHBP (53.26 ± 0.85 mg GAE/g). The FC varied between 3.09 ± 0.59 and 104.13 ± 3.80 mg QE/g, being highest in DCMMBP (104.13 ± 3.80 mg QE/g). The FC of the hexane-partitioned extracts could not be determined due to the absorption being too low (below the range of the quercetin standard curve). Since the DCM-partitioned extract of each type of BP had the significantly highest TPC and FC, they were used in the subsequent screening for antioxidant and enzyme inhibitory activities.

Antioxidant activity of the partitioned extracts

The results for the three different antioxidant assays of the DCM-partitioned extracts from the six types of BP are shown in the Table 3 along with that for ascorbic acid as the standard reference for the DPPH and ABTS assays. The EC50 values ranged from 176.85 ± 8.31 to 2,305.98 ± 59.53 µg/mL. Only the DCMMBP extract showed a significantly strong DPPH radical scavenging ability (EC50 of 176.85 ± 8.31 µg/mL), although this was still almost 2.6-fold less effective than ascorbic acid (EC50 of 68.00 ± 1.13 µg/mL), while the DCMNBP extract had the weakest DPPH scavenging capacity.

Table 2 The TPC and FC of the partitioned BP extracts.

Partitioned extract	CBP	HBP	MBP	NBP	XBP	ABP	
TPC (mg GAE/g):	
MT	29.45 ± 0.94b	20.19 ± 0.75b	24.04 ± 0.37b	11.84 ± 0.23a	18.16 ± 0.16b	26.55 ± 0.37b	
DCM	42.44 ± 0.25c	53.26 ± 0.85c	47.82 ± 0.39c	16.83 ± 0.04b	47.91 ± 0.36c	39.33 ± 0.66c	
HX	10.45 ± 0.24a	9.69 ± 0.47a	7.20 ± 0.25a	12.48 ± 0.41a	10.03 ± 0.57a	11.90 ± 0.18a	
FC (mg QE/g):	
MT	8.86 ± 0.44a	3.09 ± 0.59a	6.39 ± 0.27a	–	4.27 ± 0.16a	16.95 ± 0.44a	
DCM	31.07 ± 2.94b	56.13 ± 2.52b	104.13 ± 3.80b	6.74 ± 0.45	58.44 ± 1.36b	24.64 ± 0.44b	
HX	–	–	–	–	–	–	
Notes.

Data are shown as the mean ± 1SD derived from three replicates. Means within a column with a different superscript letter are significantly different [p < 0.05; one-way ANOVA and Post Hoc (Tukey) test (TPC) or Independent-Samples T-Test (FC)].

Table 3 Antioxidant activity of the partitioned extracts.

Sample	DPPH EC50 (µg/mL)	ABTS EC50 (µg/mL)	ABTS (µmole TE/g)	FRAP (µmole Fe2+/g)	
DCMCBP	1,952.99 ± 8.44c	563.85 ± 23.48cd	163.92 ± 10.24bc	412.11 ± 11.04c	
DCMHBP	2,291.18 ± 36.99d	681.1 ± 28.83cd	164.80 ± 7.60bc	395.17 ± 13.00bc	
DCMMBP	176.85 ± 8.31b	195.59 ± 11.44b	296.95 ± 16.87d	1,058.92 ± 28.78e	
DCMNBP	>4,000e	>2,000e	35.71 ± 2.98a	141.83 ± 5.61a	
DCMXBP	1,970 ± 3.34c	494.04 ± 25.41c	176.58 ± 10.91c	469.61 ± 15.86d	
DCMABP	2,305.98 ± 59.53d	875.04 ± 49.84d	136.22 ± 13.60b	351.97 ± 15.67b	
Ascorbic acid	68.00 ± 1.13a	44.54 ± 0.19a	–	–	
Notes.

Data are shown as the mean ± 1 SD. Within a column, EC50 means with a different superscript letter are significantly different (p < 0.05; one-way ANOVA and Post Hoc (Dunnett T3) test. Likewise, for the mean ABTS and FRAP µmole TE g−1 and µmole Fe2+ g−1 values, means within a column with a different superscript letter are significantly different [p < 0.05; one-way ANOVA and Post Hoc (Tukey) test].

For the ABTS assay, the EC50 values (Table 3) ranged from 195.59 ± 11.44 to 875.04 ± 49.84 µg/mL. Again, only DCMMBP showed a significantly strong ABTS radical scavenging ability (EC50 of 195.59 ± 11.44 µg/mL), although this was still over four-fold less effective than ascorbic acid (EC50 of 44.54 ± 0.19 µg/mL), while DCMNBP demonstrated the weakest ABTS scavenging capacity.

Furthermore, the ABTS assay was also evaluated in terms of the TEAC, determined from the Trolox calibration curve (y =  − 3.0279x + 0.5931; R2 = 0.9978). The TEAC values of the BP extracts ranged from 35.71 ± 2.98 to 296.95 ± 16.87 µmole TE/g (Table 3), where DCMMBP had the significantly highest TEAC value and DCMNBP the lowest.

The reducing power, in terms of the FRAP value, of the DCM-partitioned extracts was calculated from the calibration curve of FeSO4 (y = 0.0004x − 0.0292; R2 = 0.9947). The significantly highest antioxidant activity was found in DCMMBP (1,058.92 ± 28.78 µmole Fe2+/g) (Table 3), while the lowest was found in DCMNBP (141.83 ± 5.61 µmole Fe2+/g).

Overall, from the three antioxidant assays (DPPH, ABTS, and FRAP), DCMMBP showed the strongest antioxidant activity. However, its activity was lower than ascorbic acid (positive standard). To increase its activity, DCMMBP was further enriched (fractionated) by chromatographic methods.

Enzyme inhibitory activity

In vitro TYRI activity

Due to the large number of fractions to be screened, all the partitioned extracts were initially screened for TYRI activity at a single final extract concentration of 200 µg/mL. The obtained A475 was converted to the TYRI activity (%) and the results are presented as the mean ± SD in Table 4. At this extract concentration, DCMCBP, DCMHBP, DCMXBP, and DCMABP showed a more than 50% TYRI activity, and so they were further evaluated at different concentrations and their derived IC50 values are shown in Table 4. Of these extracts, DCMHBP had the significantly highest TYRI activity (IC50 42.63 ± 1.65 µg/mL) but this was significantly (some 4.4-fold) less effective than kojic acid (IC50 of 9.61 ± 0.47 µg/mL).

Table 4 The TYRI and LOXI activities (%/IC50) of the partitioned extracts.

Sample	TYRI (% at 200 µg/mL)/IC50	LOXI (% at 60 µg/mL)/IC50	
DCMCBP	58.88 ± 0.36/101.13 ± 3.18c	25.80 ± 4.73/–	
DCMHBP	60.41 ± 0.45/42.63 ± 1.65b	28.58 ± 4.96/–	
DCMMBP	10.05 ± 0.86/–	78.60 ± 2.81/32.55 ± 1.31b	
DCMNBP	0.00 ± 0.00/–	15.64 ± 3.28/–	
DCMXBP	52.95 ± 0.09/179.97 ± 3.77d	28.27 ± 1.82/–	
DCMABP	51.76 ± 0.93/193.87 ± 5.06e	24.47 ± 2.95/–	
Kojic acid	–/9.61 ± 0.47a	–	
NDGA	–	–/22.41 ± 2.35a	
Notes.

The IC50 values are shown as the mean ± 1SD. Within a column, means with a different superscript letter are significantly different (p < 0.05; one-way ANOVA and Post Hoc Tukey test for TYRI activity and Independent-Samples T-Test for LOXI).

In vitro LOXI activity

The partitioned extracts were also initially screened for LOXI activity at a single final extract concentration of 60 µg/mL, with the LOXI activity (%) presented as the mean ± SD in Table 4. At this extract concentration, only DCMMBP provided a high in vitro LOXI activity (78.60 ± 2.81%) and so was further evaluated at different concentrations, revealing an IC50 value of 32.55 ± 1.31 µg/mL (Table 4), some 1.4-fold less significantly effective than NDGA (22.41 ± 2.35 µg/mL).

Antioxidant and LOXI activities of compounds from DCMMBP

Fractionation of DCMMBP by SiG60-CC

Since the DCMMBP provided the best LOXI and antioxidant activities, the sample (5.64 g) was further enriched using SiG60-CC. A total of 74 fractions were collected, but after pooling fractions with a similar TLC plate profile, five different fractions (DCMMBP1–5) were obtained (Supplement 2). The TLC profile DCMMBP1 (lane 2, Rf of 0.90) revealed one major band under UV light and a few bands after staining with anisaldehyde, while DCMMBP2 (lane 3, Rf of 0.26, 0.38, and 0.59) showed two bands under UV light and one more major band after staining. For DCMMBP3 (lane 4, Rf of 0.21), it showed only one major band under UV light and so it might be a pure compound, while DCMMBP 4 (lane 5, Rf of 0.00 and 0.21) and DCMMBP 5 (lane 6, Rf of 0.00 and 0.21) each showed a major band at the base line and a minor band under UV light. When the TLC profiles of each fraction were compared to the original fraction (lane 1, Rf of 0.23), the compound in fraction 3 was found to be the main active compound. Their respective weights and characteristics are recorded in Supplement 1.

All five pooled fractions were separately screened for their antioxidant and LOXI activities. An antioxidant activity of more than 50% was found in DCMMBP3 (81.21 ± 1.38%) and DCMMBP4 (69.52 ± 1.52%) at a concentration of 500 µg/mL based on the DPPH assay. The ABTS assay revealed more 50% activity for the DCMMBP2 (92.87 ± 4.00%), DCMMBP3 (99.32 ± 0.01%), DCMMBP4 (97.97 ± 0.57%), and DCMMBP5 (72.12 ± 1.49%). However, at a concentration of 20 µg/mL, a LOXI activity of over 50% was only found in DCMMBP3 (73.35 ± 0.85%). Therefore, these fractions were further investigated at various concentrations and their respective EC50 and IC50 values derived.

The data for the DPPH assay are shown in Table 5, where the EC50 value of DCMMBP3 and DCMMBP4 was 54.66 ± 3.45 and 184.84 ± 5.47 µg/mL, respectively, and DCMMBP3 was significantly more effective than ascorbic acid (68.00 ± 1.13 µg/mL) and the parental DCMMBP extract (176.85 ± 8.31 µg/mL; Table 3).

For the ABTS assay, the obtained IC50 values are shown in Table 5. The EC50 value and µmole TE/g for the ABTS assay of DCMMBP3 was 24.56 ± 2.99 µg/mL and 2,529.69 ± 142.16 µmole TE/g, respectively, which was significantly stronger than that for ascorbic acid (44.54 ± 0.19 µg/mL) and the parental DCMMBP extract (195.59 ± 11.44 µg/mL and 296.95 ± 16.87 µmole TE/g, respectively).

For the FRAP assay (Table 5), DCMMBP3 was 3,466.17 ± 81.30 µmole Fe2+/g, which was over three-fold higher than that the parental DCMMBP extract (1,058.92 ± 28.78 µmole Fe2+/g; Table 3).

For the LOXI activity, the IC50 values are reported in Table 5. The IC50 value for the LOXI activity of DCMMBP3 was 12.11 ± 0.36 µg/mL, which was significantly (1.85- and 2.69-fold) more effective than that of NDGA (IC50 of 22.41 ± 2.35 µg/mL) and the parental DCMMBP extract (IC50 of 32.55 ± 1.31 µg/mL; Table 4), respectively. Therefore, DCMMBP3 was further enriched by HPLC.

Table 5 Antioxidant and LOXI activities of the respective fractions after SiG60 and HPLC chromatography.

Sample	DPPH EC50 (µg/mL)	ABTS EC50 (µg/mL)	ABTS (µmol TE/g)	FRAP (µmol Fe2+/g)	LOX IC50 (µg/mL)	
After SiG60-CC:	
DCMMBP1	–	–	86.63 ± 6.84a	304.50 ± 33.46a	–	
DCMMBP2	–	125.81 ± 12.97d	878.09 ± 73.48c	1,038.11 ± 34.60c	–	
DCMMBP3	54.66 ± 3.45a	24.56 ± 2.99a	2,529.69 ± 142.16e	3,466.17 ± 81.30e	12.11 ± 0.36a	
DCMMBP4	184.84 ± 5.47c	66.1 ± 3.55c	1,482.24 ± 63.00d	1,363.11 ± 47.30d	–	
DCMMBP5	–	318.66 ± 8.66e	456.82 ± 45.31b	657.56 ± 23.52b	–	
Ascorbic acid	68.00 ± 1.13b	44.54 ± 0.19b	–	–	–	
NDGA	–	–	–	–	22.41 ± 2.35b	
After HPLC:						
DCMMBP3-1	53.05 ± 2.60a	25.38 ± 0.67a	2,527.63 ± 7.99a	3,477.33 ± 52.07a	11.31 ± 0.46a	
DCMMBP3-2	57.7 ± 2.77a	28.29 ± 1.05b	2,339.16 ± 34.62b	3,091.78 ± 44.48b	11.57 ± 0.93a	
Ascorbic acid	68.00 ± 1.13b	44.54 ± 0.19c	–	–	–	
NDGA	–	–	–	–	22.41 ± 2.35b	
Notes.

Data are shown as the mean ± 1SD. Means within a column with a different superscript letter are significantly different (p < 0.05; one-way ANOVA plus for samples after SiG60-CC: Post Hoc (Tukey) test for DPPH and FRAP data, Dunnett T3 test for ABTS data, independent samples T-Test for IC50 values; for post-HPLC data: Tukey test for DPPH and ABTS EC50 data, independent samples T-Test for ABTS µmol TE/g and FRAP values; and Dunnet test for LOX IC50 values).

Fractionation of DCMMBP3 by HPLC

To optimize the separation, the HPLC was eluted with an isocratic gradient of 0:100 to 60:40 (v/v) H2O: MeOH, which separated DCMMBP3 into five peaks, with the two main peaks eluting at a retention time of 15.352 and 20.182 min, respectively, (Fig. 2). These two fractions (DCMMBP3-1 and DCMMBP3-2) were defined as compounds 1 and 2, respectively. Their weight and characters are summarized in Supplement 1. Compounds 1 and 2 were tested for their antioxidant and LOXI activities, where compounds 1 and 2 showed similar antioxidant and LOXI activities (Table 5). The structure of compounds 1 and 2 was characterized by 1H-NMR and MS analyses, and found to share the same structure, consistent with their bioactivities.

Figure 2 The HPLC chromatogram of DCMMBP3 showing the elution of DCMMBP3-1 and DCMMBP3-2 at a retention time of 15.352 and 20.182 min, respectively.

Structural identification of compounds 1 and 2

After fractionation of DCMMBP3 by HPLC, compounds 1 and 2 were obtained. The 1H-NMR spectra of compounds 1 (Supplement 3) and 2 (Supplement 4) revealed essentially similar signals to each other. Their MS spectra also showed identical pseudomolecular ions [M+Na]+ and [M+H]+ as well as mass fragment ion [M+Na-177]+ (Supplement 5–Supplement 7) at 696, 674, and 522, respectively. The above mentioned spectroscopic data suggested that compounds 1 and 2 were isomeric compounds. Compound 1 exhibited chemical shifts of the methylene hydrocarbon (-CH2-) at 1.43 and 1.87 ppm and of a methylene hydrocarbon connected to a nitrogen atom at 3.18 ppm, thus indicating the presence of a spermidine moiety. A cluster of singlet methoxy peaks at around 3.75 ppm together with the chemical shifts between 6.73 and 7.48 ppm suggested the presence of a phenolic compound containing methoxy groups. The chemical shifts at 5.95 and 6.40 ppm with small coupling constants of 12.9 Hz indicated cis-olefinic protons, while those of 6.40 and 7.40 ppm with large coupling constants of 15.8 Hz suggested trans-olefinic protons. These signals in the aromatic region suggested the presence of cis- and trans-ferulyl moieties (Fig. 3A). To determine the ratio of cis- and trans-ferulyl moieties in the whole structure, the signal integration of olefinic protons was analyzed. A ratio of 1:2 suggested that compound 1 was comprised one cis-ferulyl and two trans-ferulyls in the molecule. Therefore, three possible structures of compound 1 were established; N1, N5, N10-tri-(Z, E, E)-, N1, N5, N10-tri-(E, Z, E)-, and N1, N5, N10-tri-(E, E, Z)-feruloyl spermidine, respectively, as shown in Figs. 3C–3E. An attempt to determine exactly which structure matched to the spectroscopic data was unsuccessful due to the severely overlapped olefinic protons. In nature, the cis-isomer is less stable and gradually isomerizes into the trans-congener. Compound 2 was nearly identical to compound 1, except for the lack of cis-olefinic protons (Fig. 3B). This data suggested that compound 2 is acyl spermidine containing all trans-ferulyl moieties. Therefore, the structure of 2 was established as N1, N5, N10-tri-(E, E, E)-feruloyl spermidine, as shown in Fig. 3F.

Figure 3 1H-NMR (500 MHz, MeOD-D4) in the range of 7.6–5.7 ppm of (A) compound 1 and (B) compound 2. The chemical structures of (C–F) spermidine derivatives.

1H-NMR (500 MHz, MeOD-D4) in the range of 7.6–5.7 ppm of (A) compound 1 and (B) compound 2. The chemical structures of (C) N1,N5,N10-tri-(Z,E,E)-feruloyl spermidine, (D) N1,N5,N10-tri-(E,Z,E)-feruloyl spermidine, (E) N1,N5,N10-tri-(E,E,Z)-feruloyl spermidine, and (F) N1, N5, N10-tri-(E,E,E)-feruloyl spermidine.

DCMMBP3 fraction (Mixture of compounds 1 and 2)

1H-NMR (500 MHz, methanol-d4) δ 7.54–7.33 (m, 5H), 7.14–6.62 (m, 17H), 6.61–6.43 (m, 1H), 6.43–6.31 (m, 3H), 3.81 (qd, J = 7.5, 3.5 Hz, 20H), 3.76–3.70 (m, 3H), 3.60–3.48 (m, 4H), 3.48–3.39 (m, 2H), 3.32 (d, J = 5.8 Hz, 12H), 3.26–3.13 (m, 1H), 1.95–1.78 (m, 3H), 1.74–1.51 (m, 7H), 1.50–1.36 (m, 1H), and 1.31–1.23 (m, 1H).

Discussion

Different types of BP can provide different amounts and types of secondary metabolites, depending on their floral origin, which can result in a wide range of bioactivities between different BPs(Rzepecka-Stojko et al., 2015). In our work, six monofloral A. mellifera BPs were evaluated because many types of monofloral BP are harvested in Thailand due to the large amount of diverse monocrops that are cultivated. Moreover, the floral origin of monofloral BP is easier to identify, which makes it easier to control the quality and pharmaceutical properties of the BP compared to that of multifloral BP (Campos et al., 2010). Since bee products are edible, the food safety aspect should be considered, including the risks associated with pesticides, toxic metals, mycotoxins, allergens, and pollen grains of inedible plants (Vegh et al., 2021). There is a risk bias in certain plant groups (Helianthuus, Rosaceae, Raxinus, and Sophora) that can contain high concentrations of toxic metals (Kostic et al., 2015). In addition, BPs from pine, canola, kiwi, willow, corn poppy, rose, lotus, and camellia contain allergens that can induce anaphylaxis in some people (Yang et al., 2019).

The most practical method to determine the principal botanical origin(s) of BP is palynological analysis using microscopic examination (Kieliszek et al., 2018), but molecular analysis can also be used for minor components. Partitioning of BP with different polar organic solvents can separate the compounds based upon their polarity. This process makes further enrichment easier by screening for only the active extracts to undergo further fractionation. Of the different solvent extracts, the DCM-partitioned extracts of the six BPs in this study all showed the highest TPC and FC. Among these six DCM-partitioned BP extracts, DCMMBP and DCMHBP had the highest amount of both TPC and TFC. This result was consistent with Chantarudee et al. (2012) who reported that the DCM-partitioned extracts of corn (Zea mays) BP had the highest antioxidant activity compared to that of the MeOH- and hexane-partition extracts. Moreover, Bittencourt et al. (2015) reported that DCM partitioning of Brazilian BP enriched its antioxidant activity. As in previous reports, the active component for the antioxidant, TYRI, and LOXI activities in natural products were found to be phenolic and flavonoid compounds (Rebiai & Lanez, 2012; Sroka, Sowa & Drys, 2017; Kim et al., 2015). Therefore, the DCM-partitioned extracts of the BPs that provided the highest TPC and TFC were selected to study for their antioxidant, TYRI, and LOXI activities.

In this work, the in vitro LOXI and TYRI activities were assayed using soybean LOX and mushroom TYR since both of these enzymes show structural and functional similarities with human enzymes. Therefore, they are commonly used as a model for human enzyme inhibition (Moita et al., 2014; Chang, 2009). The highest TYRI activity was found in DCMHBP followed by in DCMMBP, while the strongest LOXI activity was found in DCMMBP. Both activities were also correlated with the very high TPC and FC found in DCMHBP and DCMMBP. It was previously reported that phenol or flavonoid compounds can chelate the copper and iron ions in the active sites of TYR and LOX enzymes, respectively, and so it is highly possible that an antioxidant compound, especially a phenolic compound, can also inhibit 5-LOX (Obaid et al., 2021; Ratnasari, Walters & Tsopmo, 2017).

By comparison to the polyamine derivatives separated from Microdesmis keayana roots (Zamble et al., 2006), the two main active components in our work are likely to be N1, N5, N10-triferuloyl spermidine or Keayanidine C. This finding agrees with the report on the phenolic profile in monofloral BP in Brazil, which also found N1, N5, N10-triferuloyl spermidine in Mimosa scabrella BP (De-Melo et al., 2018). However, in our work, the 1H-NMR of compound 1 showed cis-olefinic hydrogen and trans-olefinic hydrogen at a 1:2 ratio, while compound 2 showed trans-olefinic hydrogens. Therefore, compounds 1 and 2 are N1, N5, N10-tri(E, E, E)feruloyl spermidine and N1, N5, N10-tri(Z, Z, Z)feruloyl spermidine, respectively, which are different in their configuration. The HPLC chromatographic profile agreed with previous works that reported on the separation of this compound by HPLC from Hippesastrum × hortorum pollen (Youhnovski, Werner & Hesse, 2001) and Sambucus nigra L. (Kite et al., 2013). According to their findings, the HPLC chromatogram of this compound revealed four peaks corresponding to the four isomers: ZZZ, EZZ, ZZE, and EEE. These support that the peaks in our chromatogram are N1, N5, N10-triferuloyl spermidine in different conformations. In fact, the E- and Z-feruloyl moiety can be interconverted on exposure to UV light, in which the Z-isomer is dominant. Although compound 1 contained a small proportion of Z-feruloyl moiety, the antioxidation and LOXI activity were comparable to those of compound 2, whose structure had no Z-feruloyl residues. These observations preliminarily suggest that the geometry of C = C unsaturation contributed to a low or no effect on the biological activity. Thus, the feruloyl spermidines could potentially be applied as a mixture of isomers without the need for further time-consuming separation processes, such as HPLC, to obtain an individual pure isomer. In addition to feruloyl spermidines, other related spermidines acylated by a series of phenolics have been reported. A variety of coumaroyl spermidines isolated from rape BP (Zhang, Liu & Lu, 2020) were also demonstrated to have potent antioxidant activities in the DPPH, ABTS, and FRAP assays.

From the data mentioned above, these results showed that the BP samples harvested in Thailand, especially M. diplotricha BP, had potential bioactivities. The findings additionally indicated the relationship between the phytochemicals and functional/pharmacological application. Also, the obtained data supported the traditional or alternative use of M. diplotricha BP in the indigenous medicine of Thailand.

Conclusions

Mimosa diplotricha BP had the highest antioxidant and LOXI activities, which were correlated to the highest TPC and FC. Although both bioactivities of the DCM-partitioned BP extracts were detected, their fractionation led to an enrichment of triferuloyl spermidines and enhanced specific bioactivities, suggesting these might be the major active compounds in the Mimosa diplotricha BP. In addition, this work supported the importance of the floral origin of BPs, since it affected the bioactivity directly. Here, while Mimosa diplotricha BP had the best antioxidant and LOXI bioactivities, Helianthus annuus L. BP had the best TYRI activity.

Supplemental Information

Supplemental Information 1 The weight, yield, and the character of the different bee pollen (BP) extracts

Click here for additional data file.

Supplemental Information 2 Representative thin layer chromatography (TLC) images showing the compound profile of the original dichloromethane-partitioned BP extracts of M. diplotricha or DCMMBP (lane 1), and its subfractions DCMMBP1 (lane 2), DCMMBP2 (lane 3), DCMMBP3 (lane 4), DCM

The mobile phase was 7% MeOH-DCM.

Click here for additional data file.

Supplemental Information 3 Mass fragment of compound 1 and compound 2 in positive mode

Click here for additional data file.

Supplemental Information 4 Proton nuclear magnetic resonance (1H-NMR) peak data at chemical shift(δ) 0.0–8.5 ppm in length of DCMMBP3-1 fraction or compound 1 in deuterated methanol (MeOD-D4)

Click here for additional data file.

Supplemental Information 5 1H-NMR peak data at chemical shift(δ) 0.0–8.5 ppm in length of DCMMBP3-2 fraction or compound 2 in MeOD-D4

Click here for additional data file.

Supplemental Information 6 Mass spectrum of DCMMBP3-1 fraction or compound 1 at m/z 500–1,000

Click here for additional data file.

Supplemental Information 7 Mass spectrum of DCMMBP3-2 fraction or compound 2 at m/z 500–1,000

Click here for additional data file.

Additional Information and Declarations

Competing Interests

Author Contributions

Data Availability

The authors declare there are no competing interests.

Phanthiwa Khongkarat conceived and designed the experiments, performed the experiments, analyzed the data, prepared figures and/or tables, authored or reviewed drafts of the article, and approved the final draft.

Preecha Phuwapraisirisan conceived and designed the experiments, performed the experiments, analyzed the data, prepared figures and/or tables, authored or reviewed drafts of the article, and approved the final draft.

Chanpen Chanchao conceived and designed the experiments, performed the experiments, analyzed the data, prepared figures and/or tables, authored or reviewed drafts of the article, and approved the final draft.

The following information was supplied regarding data availability:

The raw data is available in the Supplemental Files.

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
