# Peer review of "Phytochemical content, especially spermidine derivatives, presenting antioxidant and antilipoxygenase activities in Thai bee pollens"

_PeerJ, doi:10.7717/peerj.13506_

## Round 0.1 · original submission · Major Revisions

The manuscript needs substantial revision, additional work, and justifications in order to appreciate the quality for publication. Reviewers have commented against the acceptance of the manuscript in its current form as it suffers from serious concerns regarding the implemented protocol as well as the presentation of the data. Moreover, thorough English editing is required. Please revise the manuscript taking help from a colleague who is proficient in English and familiar with the subject matter, who can review your manuscript, or contact a professional editing service to review your manuscript. Revise and resubmit accordingly.

Reviewer 1 ·

Basic reporting

The paper aims to determine the total phenolic and flavonoid contents, antioxidant activities in simple non-cellular tests (DPPH, ABTS, TEAC, and FRAP assays), and anti-tyrosinase and anti-lipoxygenase activities of fractionated extracts of hexane, dichloromethane, and methanol obtained from six monofloral Apis mellifera bee pollen, i.e. Camellia sinensis, Mimosa diplotricha, Helianthus annuus, Nelumbo nucifera, Xyris complanate, and Ageratum conyzoides. The manuscript is well organized and meets the criteria of a professional article structure. The conclusions are consistent with the results obtained and with the data already existing in the literature. Literature references are also up-to-date and selected to provide sufficient background.

There is no explanation in the introduction section as to why bee pollen from these six plant species was selected for research.

The discussion also lacks an explanation of the potential application, including medicinal use, of the extracts tested in the manuscript.

Experimental design

The idea of the work seems to be very interesting as bee pollens are a rich source of many nutrients, including proteins, and are widely used as a natural dietary supplement. Therefore, the subject of the manuscript most closely corresponds to the journal's aims and scopes. The materials and methods are described quite thoroughly, although, in the case of the place of origin of the samples, an additional table must be added to the manuscript.

L123-124: Please add a table for all tested samples with the exact location (longitude and latitude) of the sample collection site and the exact time of collection.

Additionally, the manuscript contains far too many tables and figures. The Authors should only leave Figures 1, 6, 8 (Fig. 3 should be transferred to the supplements); Tables 2-5 (Table 1 should be transferred to the supplements), and Supplements 8-12.

Figures: 2, 4, 5, 7, and Supplements: 1-7 should be removed, as they present the same results presented only in a different form.

Minor issues:
L1-4: Please change the title of the manuscript as it suggests testing only bee pollen from one plant.
L395-403: Please use the Rf values in the description of the TLC plate profiles of five different fractions (DCMMBP1-5).
L360-364: Why was ascorbic acid activity (positive standard) not tested in the FRAP test?
L512: in vitro should be in italics.
L632: Hippesastrum x hortorum should be in italics.

Validity of the findings

In my opinion, the submission requires major editions and improvement at a few points, the main of which are listed above.

Reviewer 2 ·

Basic reporting

No comment.

Experimental design

No comment.

Validity of the findings

No comment.

Additional comments

Thanks for giving me opportunity to review this manuscript. The manuscript entitled, "Phytochemical content, especially spermidine derivatives, presenting antioxidant and anti-lipoxygenase activities in Mimosa diplotricha bee pollen" has been well written and nicely presented by the authors.

I would like to recommend this article for acceptance in your journal after minor correction.
My suggestion is if you want to add Fig. 2, you should add data of all samples along with control ascorbic acid in different concentrations with all assays, otherwise you can remove Fig. 2.

·

Basic reporting

no comment

Experimental design

no comment

Validity of the findings

no comment

Additional comments

The manuscript titled “Phytochemical content, especially spermidine derivatives, presenting antioxidant and antilipoxygenase activities in Mimosa diplotricha bee pollen” by Khongkarat and others describes a detailed phytochemical analysis of six A. mellifera BP samples with bioguided isolation of triferuloyl spermidines. This is a well-planned study and the results obtained are interesting. I have the following comments for the authors to address.
1. Revised the title. The present title is not informative.
2. In the abstract, reduce the usage of obtained numerical values.
3. Write a reference for the sentence “In Thailand, some dominant floral origins are different from those 109 in other countries” in the introduction section.
4. Abbreviations must be defined at first use in the text. There is excessive use of abbreviations particularly in the Tables and Figures. These must be defined in their respective locations.
5. The results should be written keeping in view the statistical summaries of the data.
6. There is similarity of data presented here with the manuscript published by same authors in the same journal (PeerJ 10:e12722). Repetition of data must be avoided.
7. Each supplemental file must have proper titles and description. Also define the abbreviations used in these supplemental files.
8. The English language should be improved.

---

## Round 0.2 · Minor Revisions

The manuscript is significantly improved by the authors. However, I can still see some language and typo errors in the manuscript. I strongly recommend the authors thoroughly recheck the manuscript or consider professional editing and resubmit the revised version.

Reviewer 1 ·

Basic reporting

-

Experimental design

-

Validity of the findings

-

Additional comments

The manuscript has been revised in accordance with the Reviewers' recommendations and is now suitable for publication in PeerJ.

---

## Round 0.3 · accepted · Accept

Manuscript is significantly improved by the authors and now can be accepted in its current form.